# Diet and Healthy Lifestyle in the Management of Gestational Diabetes Mellitus

**DOI:** 10.3390/nu12103050

**Published:** 2020-10-06

**Authors:** Louise Rasmussen, Charlotte Wolff Poulsen, Ulla Kampmann, Stine Bech Smedegaard, Per Glud Ovesen, Jens Fuglsang

**Affiliations:** 1Department of Obstetrics and Gynaecology, Aarhus University Hospital, Palle Juul-Jensens Boulevard 99, 8200 Aarhus N, Denmark; loura7@rm.dk (L.R.); charpoul@rm.dk (C.W.P.); per.ovesen@clin.au.dk (P.G.O.); 2Steno Diabetes Center Aarhus, Aarhus University Hospital, Hedeager 3, 8200 Aarhus N, Denmark; ullaopst@rm.dk (U.K.); stinsmed@rm.dk (S.B.S.)

**Keywords:** gestational diabetes mellitus, GDM, pregnancy, lifestyle, diet, nutrition, weight management, physical activity

## Abstract

Gestational diabetes mellitus (GDM) among pregnant women increases the risk of both short-term and long-term complications, such as birth complications, babies large for gestational age (LGA), and type 2 diabetes in both mother and offspring. Lifestyle changes are essential in the management of GDM. In this review, we seek to provide an overview of the lifestyle changes which can be recommended in the management of GDM. The diet recommended for women with GDM should contain sufficient macronutrients and micronutrients to support the growth of the foetus and, at the same time, limit postprandial glucose excursions and encourage appropriate maternal gestational weight gain. Blood glucose excursions and hyperglycaemic episodes depend on carbohydrate-intake. Therefore, nutritional counselling should focus on the type, amount, and distribution of carbohydrates in the diet. Further, physical activity has beneficial effects on glucose and insulin levels and it can contribute to a better glycaemic control.

## 1. Introduction

Pregnant women gradually develop insulin resistance during pregnancy, thereby ensuring sufficient nutrient supply for the growing foetus [1]. In women with gestational diabetes mellitus (GDM), the insulin resistance leads to hyperglycaemia [2,3]. The definition of GDM is glucose intolerance with onset or first recognition during pregnancy [3]. Glucose passes through the placenta to the foetus and increases foetal insulin production, which, in turn, stimulates foetal growth, causing macrosomia and children large for gestational age (LGA) [4]. In the short-term, GDM is associated with increased risk of adverse pregnancy outcomes with a following long-term risk of childhood obesity and type 2 diabetes in mother and offspring [5]. The prevalence of GDM is rising [4], and so is the need for treatment. 

Lifestyle changes are essential in the management of gestational diabetes. First-line treatment in GDM is medical nutrition therapy, together with weight management and physical activity [6,7]. It has been suggested that lifestyle modification alone is sufficient to control blood glucose in 70–85% of the women that were diagnosed with GDM [7]. How the diet should be composed for women with GDM is a complex matter and still not completely settled. In this review, we seek to provide an overview of the most important dietary interventions and components and how to treat and guide each woman with GDM during pregnancy. 

## 2. Energy Requirements

### 2.1. Optimal Weight Gain

The recommended weight gain during pregnancy in women with GDM is the same when considering normal glucose tolerance pregnancies (NGTP). Gestational weight gain (GWG) should maintain the growth and development of the foetus [8]. The weight recommendations vary slightly from country-to-country. However, many countries refer to the recommendations for GWG that were made in 1990 by the Institute of Medicine (IOM) of National Academies, which were updated in 2009 based on pre-pregnancy Body Mass Index (BMI) (See Table 1) [8]. 

These weight gain guidelines are based on studies that indicated that women, whose weight gains are outside the recommended ranges, are at increased risk of adverse maternal and neonatal outcomes, such as pregnancy complications, maternal postpartum weight retention, and obesity, in the offspring [9].

In the guidance of pregnant women, a recommended rate of weight gain during 2nd and 3rd trimester can be helpful. Hence, women with a BMI of less than 18.5 kg/m^2^ should be recommended a weight gain between 0.44–0.58 kg/week. Women with a BMI between 18.5 to 24.9 kg/m^2^ should be recommended a weight gain between 0.35–0.50 kg/week. Women with a BMI between 25.0 to 29.9 kg/m^2^ should be recommended a weight gain between 0.23–0.33 kg/week and, finally, women with a BMI of 30 kg/m^2^ or above should be recommended a weight gain between 0.17–0.27 kg/week [8].

### 2.2. Energy Requirements for Normal or Underweight Women

There is not sufficient evidence to suggest that the energy requirements for women with GDM should be different from normoglycemic women or suggest a specific optimal calorie intake for women with GDM [10]. In the clinic, the energy expenditure can be calculated using the equation by Henry multiplied by a factor of physical activity level (PAL) or the equations that were recommended by the IOM (see Table 2).

The physical activity coefficient or level (PA/PAL) can be determined by the reference values that were given by the Nordic Nutrition Recommendations (NNR) or the IOM (see Table 3).

An additional assessment of daily energy requirements during pregnancy is based on trimesters, although there is no international agreement on the exact calorie requirements during the three trimesters (see Table 4). There may be considerable variance in the total energy requirement among women with GDM as in NGTP [12], and each patient should be regularly weighed during pregnancy.

### 2.3. Energy Requirements for Women with Overweight or with Excessive Gestational Weight Gain

In women with GDM, excessive weight gain has been associated with an increased risk of hypertensive disorders of pregnancy, caesarean section, and LGA-babies [13,14]. Additionally, a meta-analysis concludes that it is extremely important to prevent excessive weight gain in GDM pregnancies [14].

In women with GDM, who have already accomplished a recommended weight gain, weight stabilization is the goal and calorie restriction can be necessary. In women with obesity and GDM, a 30–33% calorie restriction has been shown to reduce hyperglycaemia and plasma triglyceride levels [15]. In a retrospective cohort by Kurtzhals et al., the women with GDM who had the best dietary adherence to an energy-restricted “diabetes diet” and the lowest weight gain had lower foetal growth (infants with a birth weight-SD (standard deviation) score of 0.15 ± 1.1 in contrast to a birth weight-SD score of 0.59 ± 1.6) and decreased HbA1c, as compared to women with GDM with the highest GWG and poor dietary adherence [5].

### 2.4. Summary, Energy Requirements

The general recommendations for weight gain and the calculation of energy requirements for NGTP are also appropriate for women with GDM. Furthermore, particular attention should be given in order to avoid excessive weight gain. In women with obesity, or women who have already reached the recommended weight gain, a calorie restriction of 30–33% may be advisable.

## 3. Carbohydrates

In women with GDM, carbohydrates are the most important macronutrient. The digestion and absorption of carbohydrates cause an increase in blood glucose levels, and postprandial hyperglycaemia is primarily dependent on carbohydrate-intake [16]. The amount and the type of carbohydrate will both impact glucose levels [7]. Thus, a high intake of carbohydrate in a meal can result in hyperglycaemia [16]. However, glucose is the principal energy substrate for the placenta and foetus, which is essential for normal foetal growth and metabolism [17]. The IOM recommends 46–65 Energy percent (E%) from carbohydrates and a minimum of 175 g of carbohydrate daily to ensure appropriate foetal growth and cerebral development and function [2,8,10]. Ketonemia and/or ketonuria should be avoided, as it has been associated with lower mental or motor function in the offspring [2]. Carbohydrates should predominantly consist of starchy foods, a low glycaemic index, and a naturally high content of dietary fibre, such as vegetables, legumes, fruits, and whole grains [2,18,19]. The intake of added sugars should be kept low. The IOM has not set a daily intake of added sugars that individuals should aim for, but recommends that the intake of added sugar is limited to no more than 25% of total energy during pregnancy [8].

### 3.1. Low-Carbohydrate Diets

There is no international agreement on an appropriate amount of daily carbohydrate intake for women with GDM. Some guidelines recommend that the daily carbohydrate intake should not exceed 40–50E% [20]. Other countries, like Denmark, follow the general recommendation for NGTP, which, in the Nordic countries, is 45–60E% [11]. Only few clinical trials comparing low-carbohydrate diets with higher-carbohydrate diets have been conducted. Hernandez et al. compared a 40% carbohydrate diet with a 60% carbohydrate diet in a randomized crossover study. The 60% carbohydrate diet consisted of higher-complex carbohydrate. The low-carbohydrate diet resulted in a lower postprandial glucose, lower daytime mean glucose concentrations, lower area under the curve of 2 h postprandial glucose, and lower 24 h total glucose area under the curve, when compared with the 60% carbohydrate diet [21]. However, in the group receiving a 60% carbohydrate diet, the postprandial glucose values were still below current targets: 1 h <140 mg/dL (7.8 mmol/L) and 2 h <120 mg/dL (6.7 mmol/L). No differences for fasting blood glucose was found [21]. Moreno-Castilla et al. did not find any differences between groups in insulin treatment or in pregnancy outcomes, such as caesarean sections, LGA-babies, macrosomia, or gestational age at delivery, when comparing a 40% carbohydrate-diet with a 55% carbohydrate diet in a non-crossover randomized study [3]. Thus, there are conflicting results and it should be pointed out that a lower carbohydrate intake will often lead to an increased intake of fat, which, outside pregnancy, has been associated with an increase in serum fatty acids, insulin resistance, and increased foetal fat accretion and infant adiposity in NGTP [21].

### 3.2. Dietary Fibres

Normally, simple carbohydrates result in higher postprandial excursions than complex carbohydrates. NNR recommends a minimum of 25 g dietary fibre for women in general [11], while the American Diabetes Association recommends a minimum of 28 g of fibre to women with GDM [10], which is similar to IOM recommendations for normoglycemic women during pregnancy [8]. These recommendations can be met by eating 600 g of fruit and vegetables a day with a minimum of 300 g vegetables, with focus on rough and fibrous vegetables and by choosing wholemeal bread, pasta, and rice.

### 3.3. Low Glycaemic Index Diets

Carbohydrate food can be classified in relation to its effect on postprandial blood glucose expressed as a percentage of the blood glucose response of a reference food (e.g., glucose solution or white bread). The Glycaemic Index (GI) is a number from 0 to 100 that is assigned to a food, with pure glucose being arbitrarily assigned the value of 100, which represents the relative rise in the blood glucose level two hours after consumption [22].

Fast absorbable carbohydrates with a GI >70 are considered as high GI foods, while slowly absorbed carbohydrates with a GI ≤55 are considered low GI foods [22]. Moses et al. did show a reduced need for insulin in women with GDM, when they consumed a diet with a low GI in a RCT of 63 women with GDM. Even though Moses et al. compared with a diet high in fibre and a low sugar content, a lower GI diet significantly reduce insulin requirements in women with GDM [23]. In a meta-analysis of six RCTs and 532 women with GDM, Xu et al. found that a low-GI diet significantly reduced 2 h postprandial glucose concentrations, without any effect on fasting plasma glucose (FPG), birth weight, HbA1c, macrosomia, or insulin requirements [24]. Moreover, in a recent systematic Cochrane review that included 19 randomized trials and 1389 women with GDM, no effect of a low GI-diet on LGA or other primary neonatal outcomes was found [25].

In the case of GI, the amount of carbohydrate is not considered, which is also a strong factor in the prediction of the postprandial blood glucose response. Glycaemic load (GL), on the other hand, is the product of the total available carbohydrate content in a given amount of food and a given GI [22]. Low GL diet has been shown to improve glycaemic control in type 2 diabetes [26]. The results might also apply to GDM, as GDM and type 2 diabetes mellitus (T2DM) are both characterized by insulin resistance [27]. In a study by Bao et al. of healthy adults, the GL was a more powerful predictor of postprandial glycaemia and insulinemia when compared to the carbohydrate content [28]. In a recent study by Lv et al., 134 women with GDM were randomly allocated to either conventional nutritional nursing or specific nutritional nursing intervention based on GL. Significant differences in fasting blood glucose and the 2 h postprandial glucose levels between the two groups was found with lower levels in the group receiving intervention based on GL [29]. No statistically significant differences in the rates of adverse pregnancy outcomes, such as preterm delivery, foetal macrosomia, and foetal distress, was found; however, there was a lower incidence of premature delivery, eclampsia, pregnancy hypertension syndrome, and foetal macrosomia in the group receiving nutritional nursing based on GL [29].

### 3.4. Meal Frequency and Carbohydrate Distribution

A daily meal frequency of three main meals and 2–3 small meals or snacks is recommended to avoid excessive food intake at the same time, more specifically to avoid large quantities of carbohydrate and, thereby, reduce the postprandial blood glucose that is illustrated in Figure 1 [2,4,20,30].

It has been suggested that breakfast should only contain small amounts of slowly absorbed carbohydrates, because there is usually a higher postprandial increase in blood glucose in the morning [20]; some guidelines recommend a maximum of 30 g carbohydrate at breakfast [30]. However, these recommendations are primarily based on personal experience and the scientific evidence is limited. In a randomized crossover study with 12 women with GDM, Rasmussen et al. demonstrated a significantly lower mean glucose and fasting blood glucose on a diet with a high carbohydrate intake in the morning as compared with a low carbohydrate intake in the morning. During both intervention periods (high and low carbohydrate in the morning), the recommended total carbohydrate intake was 46E% ± 2E%. In the same study, insulin resistance (as measured by homeostatic model assessment for insulin resistance (HOMA-IR)) significantly decreased during the period with the high carbohydrate intake in the morning. However, Rasmussen et al. also found a higher mean amplitude of glucose excursions and coefficient of variation in the group receiving a high carbohydrate intake in the morning as compared with the low intake [31]. There is a lack of randomized clinical trials studying whether a high or low carbohydrate intake in the morning is preferential.

### 3.5. Artificial Sweeteners

In the United States, the intake of artificial sweeteners (AS) during pregnancy has been increasing in recent years [32] and, in a study from Norway, it is reported that more than 40% of the pregnant women consumed artificially sweetened beverages (ASB) more frequently than once per week in early pregnancy [33]. It is conceivable that the intake of AS is particularly high in women with GDM, seeking to limit the intake of sugar and, to a greater extent, opt for “sugar-free” products and “No added sugar” products.

The Acceptable Daily Intake is defined as an estimate of the amount of food additive that can be ingested daily over a lifetime without health risk. The average use of AS, also called Non-Nutritive sweeteners (NNS), is usually below this limit and the US Food and Drug Administration and European Food Safety Authority, which regulates AS and NNS, has reported asulfame potassium, aspartame, saccharin, and steviol glycosides to be safe for use by the general public, including in pregnancy [34,35]. Observational human studies regarding AS and NNS exposure are often difficult to interpret because of heterogeneity and the lack of accuracy of self-reported intake of AS and NNS. In NGTP, some issues of concern, including increased infant BMI, childhood obesity, and small increase in preterm birth, have been observed [36]. Concerning preterm birth, the European Food Safety Authority has concluded that there is no evidence available to support a causal relationship between the consumption of ASBs and preterm delivery [37].

In a prospective study from the Danish National Birth Cohort, it was shown that approximately half of the women with GDM reported consuming ASB during pregnancy and 9% consumed it daily. When compared to no consumption, daily ASB intake during pregnancy was positively associated with an 1.57-fold increase in LGA risk in offspring, positively associated with an 0.59 SD increase in BMI z-scores at seven years and a 1.93-fold increased risk of overweight/obesity at seven years. The substitution of ASBs with water during pregnancy was associated with a 17% reduced risk for overweight/obesity at seven years, whereas sugar-sweetened beverages (SSB) substitution with ASBs was not related to a lower risk, but with an 1.14-fold increased risk of offspring overweight at seven years [38].

More studies, especially RCTs, on ASB and data with longer follow-up time are wanted.

### 3.6. Summary, Carbohydrates

Carbohydrate is the macronutrient that has the greatest impact on postprandial hyperglycaemia. Despite some studies suggesting a beneficial effect of low-carbohydrate diets, there is currently no evidence to recommend a carbohydrate intake that is lower than in NGTP and a minimum of 175 g of carbohydrate should be ensured. The exact amount of carbohydrate should be individualized, and the focus should be on the types of carbohydrate. Carbohydrates should predominantly consist of starchy foods with a naturally high content of dietary fibre, such as vegetables, legumes, fruits, and whole grains. Furthermore, carbohydrate intake should be distributed throughout the day in order to avoid excessive amounts that result in postprandial hyperglycaemia.

## 4. Protein

During pregnancy, there is an increased requirement of protein due to its role in the synthesis of maternal (blood, uterus, and breasts), foetal, and placental tissues [11]. The recommended amount of protein in the dietary treatment of GDM is similar to the general nutrition advice for normal pregnancies. The IOM recommends 10–35E% from protein during pregnancy, and an estimated average requirement of 0.88 g/kg/d with a minimum recommended daily intake of 71 g protein [8]. NNR recommends a protein intake of 10–20E% for non-pregnant adult women, corresponding to approximately 0.8–1.5 g protein/kg/d based on a PAL of 1.6 for an intake of 10E% and a PAL of 1.4, for an intake of about 20E%, respectively. Further, NNR recommends an additional safe intake of protein for healthy women during pregnancy gaining 13.8 kg of 0.7, 9.6, and 31.2 g/d during first, second, and third trimester, respectively [11]. In general, most pregnant women are able to cover their protein needs, as the increased requirement of protein is met by consuming a normal diet in a quantity that allows a weight gain within the recommended limits [11].

### 4.1. Protein Metabolism in GDM

The antepartum loss of nitrogen is lower than the postpartum loss, which suggests a reduction in protein catabolism to accrete more nitrogen to support maternal and foetal growth [39]. The loss of nitrogen is similar in GDM pregnancies and normal pregnancies [39,40]. In early GDM, when patients have less metabolic decompensation, there appears to be no difference in leucine kinetics/rate of protein turnover [41]. Later in gestation, when insulin resistance is more pronounced and antidiabetic treatment may be intensified with diet and sometimes insulin, the rate of protein turnover is increased in women with insulin treated GDM [40]. The increased protein breakdown, together with the normal urea excretion, suggests an increased pool of amino acids (AA) available to the placenta and thereby the foetus. The increased pool of AA in GDM and the association with macrosomia is unclear, as the results are often conflicting. One study found no correlation between AA and birth weight in GDM [40]; another found a correlation between leucine and birth weight for both GDM and NGTP controls [41].

### 4.2. Protein, the Placenta and GDM

A study in Chinese women with GDM found an inverse relationship between protein intake and placental size without any association with birth weight [42]. AA are carried across the placenta through an active transport system providing a greater concentration of AA in the foetus when compared to the mother [43]. In GDM, the transfer of AA across the placenta has been shown to be both decreased [44], unchanged [45], and increased [46]. A study showed elevated levels of branch chained amino acids (BCAA) in GDM as compared to pregnant women with normal glucose tolerance [47]. It has been suggested that the flux of insulinotropic AA (e.g., BCAA) over the placenta affects the beta cell of the foetus creating hyperinsulinemia affecting foetal growth [48]. Studies using metabolomics on cord blood, including both normal and GDM pregnancies, found no association between BCAA and increased insulin/c-peptide levels, thus not supporting BCAA as a cause of foetal hyperinsulinemia [49,50]. However, there was an association with birth weight, but not with the sum of skinfolds [49] or infants being LGA [50]. These findings suggest an association with lean body mass, but not with fat mass.

### 4.3. Plant vs. Animal Protein

Animal proteins are considered to be complete proteins, as they contain all nine essential AA while plant proteins are considered incomplete, as they can be deficient of one or more essential AA. However, a variety of plant based proteins consumed throughout the day provide sufficient essential AA [51]. A review including studies on vegetarian and vegan diets during pregnancies with sufficient energy and protein supply in the setting of no financial constraint concluded that vegetarian and vegan diets were safe during pregnancy if supplemented with iron and B12 [52]. However, vegans should plan their diets well, as they have an increased risk of not consuming enough protein when compared to omnivores and vegetarians [53]. An Australian study compared vegetarian and non-vegetarian women with GDM from South Asia in Australia found that the vegetarian GDM group received 14 ± 3% of their energy intake from protein as compared to 17 ± 4% in non-vegetarians, but remained within the range of the non-vegetarians supporting the feasibility of a vegetarian diet [54]. Another meta-analysis found that, overall, a vegetarian diet was not associated with birth weight, but that Asian women had a higher risk of delivering babies with low birth weight when compared to Caucasian women [55]. In poor rural areas of Asia, living a life as a vegetarian is more often a result of low income than a choice of lifestyle and lack of micronutrients e.g., vitamin B12 [56] may explain the association between vegetarianism and low birth weight.

A randomized clinical trial (RCT) of animal vs. soy protein applied for six weeks in 68 women with GDM showed lower fasting glucose, lower insulin levels, lower HOMA-IR, and lower triglyceride levels in the plant protein group. The women were randomized to receive protein from either 70% animal or 70% plant protein (half being from soy protein)—both arms were identical in the amount of protein received [57]. Another RCT on soy protein-based protein rich diet vs. high fibre complex diet in GDM showed a reduction in the need for exogenous insulin in the soy diet group. The arms of treatment were isocaloric. However, a low GI diet might explain the results rather than the protein itself [58].

### 4.4. High Protein Supplementation

Only one study on high protein supplementation during pregnancy has been performed. A RCT was performed in 1980 in poor African American women at risk of having infants with low birth weight. The high protein content of the supplementation (74.2 g/day) was associated with very early premature births, neonatal deaths, and growth retardation up to 37 weeks of gestational age [59]. It is unclear whether the adverse effects occurred because of the study population being unaccustomed to the high protein supplementation or if the results would have been different in populations of normal weight, well-nourished women, and women with GDM. However, the results of the study and lack of other studies of high protein intake during pregnancy implies that one should be reluctant regarding diets exceeding the recommend intake of protein during pregnancy—NGTP or diabetic pregnancies.

### 4.5. Pre-Meals and GDM

Pre-meals of protein administered prior to a meal have shown promising results on the postprandial blood glucose in non-pregnant healthy individuals and individuals with T2DM [60,61]. In a RCT of 52 women with GDM receiving either 8.5 g of casein hydrolysate (*n* = 26) or placebo (*n* = 26) prior to breakfast and dinner for eight days, the average blood glucose was decreased in the casein group [62]. Milk protein consists of 80% casein and 20% whey. Pre-meal whey protein has shown promising results with lower postprandial blood glucose in both healthy subjects, subjects with metabolic syndrome, and T2DM [60,61,63,64]. T2DM and GDM share similarities in their pathophysiology and, hence, women with GDM may display the same beneficial effect of whey pre-meals on blood glucose.

### 4.6. Summary, Protein

The current evidence suggests that increased protein intake from plants, lean meat and fish, and reduced intake of red and processed meat are beneficial in the treatment of GDM and may improve insulin sensitivity. The beneficial effect of plant protein on GDM might not be directly attributable to the source of protein, but rather to the reduction of other nutrients that are associated with an increased risk of GDM, such as carbohydrate [65] and saturated fat [66]. Furthermore, the results might not be generalizable to all ethnicities, as the majority of studies only investigated Asian and Middle Eastern women.

## 5. Fat

The recommended amount of fat in the dietary treatment of GDM is similar to the general nutrition advice for NGTP. The IOM recommends 20–35E% from fat [8], while the recommendation by NNR is the same as in non-pregnancy; 25–40E% [11]. A high intake of fat should be avoided, because this has been associated with infant adiposity, increased maternal inflammation and oxidative stress, and impaired muscle glucose uptake. Further, high fat diets might cause placental dysfunction [21].

### 5.1. Saturated Fatty Acids

The IOM recommends keeping the intake of trans fatty acids and saturated fatty acids as low as possible while consuming a nutritionally adequate diet during pregnancy [8]. NNR recommends, in general, that adults intake of saturated fat should not exceed 10E% [11]. To meet these recommendations, women with GDM can be instructed in choosing meat and meat products with a maximum of 10% fat, to choose low-fat dairy products, including choosing sour milk products with a maximum of 1.5% fat and limit intake of fatty dairy products, such as cream and butter.

### 5.2. Monounsaturated Fatty Acids

The recommendation for Cis-Monounsaturated fatty acids (MUFAs) by NNR is the same as in non-pregnancy; 10–20E%. In a study by Lauszus et al., 27 women with GDM were randomized to either high-carbohydrate diet or a high-MUFA diet. The 24 h diastolic blood pressure increased more in the carbohydrate group than in the MUFA-diet group. However, Lauszus et al. also found a significant difference in the intervention effect on insulin sensitivity in delta changes between groups, with a 15% increased insulin sensitivity in the high-carbohydrate diet and 34% decrease in the high-MUFA-diet [67]. More studies are needed if the recommendation for MUFA is to be changed in GDM as compared to a NGTP.

### 5.3. Polyunsaturated Fatty Acids

Long-chain polyunsaturated fatty acids (PUFAs) of the *n*-3 (α-linolenic acid) and *n*-6 series (linoleic acid) are the most important fatty acids for foetal growth and development [68,69]. *n*-3 and *n*-6 serve as essential components of cell membranes. Additionally, they are precursors for the synthesis of eicosanoids, which are important in the development of foetal nervous, immune, visual, and vascular systems [70,71,72]. The depletion of long-chain PUFAs in foetal tissues are associated with behavioural, cognitive, and visual abnormalities later in life in NGTP [68]. Furthermore, low levels of *n*-3 and *n*-6 during pregnancy have been shown to be correlated with preterm birth or foetal growth retardation in NGTP [73]. NNR recommends 5–10E% from PUFAs and a minimum of 1E% *n*-3 fatty acids in general for adults. A total intake of 2.7 g/day *n*-3 is considered to be safe during pregnancy [11].The IOM recommends 5–10E% *n*-6 and 0.6–1.2 E% *n*-3 with a minimum of 13 g/d of *n*-6 and a minimum of 1.4 g/day of *n*-3 during pregnancy [8]. An intake of a minimum of 350 g of fish per week, of which 200 g should be fatty fish, will ensure that the patients follow these recommendations. However, pregnant women should avoid predatory fish, due to the content of heavy metals, and salmon from the Baltic sea, due to pollution [74].

With regard to supplements with PUFAs, the evidence is not clear, as studies have shown conflicting results. These are plausibly reflecting the nature of long-chain PUFAs ingested, type of supplement, dose, and on the outcome evaluation. However, some studies with fish oil supplements have shown positive results in women with GDM. In an RCT by Jamilian et al., women with GDM were randomized to either 1000 mg omega-3 acids from flaxseed oil plus 400 IU vitamin E supplements or placebo for six weeks. A positive effect on biomarkers of oxidative stress and inflammation was found together with a significant rise in the total antioxidant capacity, nitric oxide, a significant decrease in plasma malondialdehyde, and a lower incidence of hyperbilirubinemia in new-borns. There was no effect on new-born outcomes (e.g., caesarean section, preterm delivery, or macrosomia >4000g) or C-reactive protein levels [75]. In another RCT by Jamilian et al., 40 women with GDM were randomly allocated to either 1000 mg fish oil capsules or placebo twice a day for six weeks. Fish oil capsules improved gene expression that was related to insulin, lipids, and inflammation; proliferator-activated receptor gamma was upregulated, and low-density lipoprotein receptor, Interleukin-1, and tumor necrosis factor alpha were downregulated. Fish oil supplement, as compared to placebo, also led to a significant reduction in FPG, serum triglycerides, and a significant increase in LDL- and HDL-cholesterol levels. Further, a significant reduction in high-sensitivity C-reactive protein, in those who received fish oil supplements, was found. However, Jamilian et al. did not find any effect on serum insulin, total cholesterol levels, or HOMA-IR [76]. In a study conducted by Samimi et al., a significant difference in changes in serum insulin and HOMA-IR was found in those women with GDM, who received fish oil supplements when compared to placebo. However, Samimi et al. did not find any effect on FPG [77]. Contrary to these results, a systematic review from 2016 did not find any effect of fish oil supplements on FPG, Homeostatic model assessment-Beta cell function, or lipid profiles. It was concluded that there is not enough evidence to support the routine use of fish oil supplements during pregnancy in the treatment of diabetes [78].

### 5.4. Summary, Fat

Women with GDM can be recommended an intake of 20–35E% from fat. The intake of saturated fat should be limited, and special focus should be placed on ensuring a sufficient intake of *n*-3 fatty acids. Despite some studies reporting a positive effect of fish oil supplementation, there are still conflicting results and, based on the current evidence, routine supplements of fish oil cannot be recommended or refuted, whereas women with GDM are recommended an intake of 350 g/week of fish as in NGTP.

## 6. Vitamins, Minerals and Tracers

During pregnancy, the need for vitamins and minerals increases [8,11,79]. There is not sufficient evidence to suggest that vitamin and mineral requirements for women with GDM should be different from normoglycaemic women or to suggest a specific optimal vitamins and minerals intake for women with GDM.

Well-nourished women may not need multiple-micronutrient supplements to satisfy daily requirements, but individual adjustments should be made upon the women’s specific needs. If pregnant women do not consume an adequate diet, then the IOM recommends multiple-micronutrient supplements [80]. As a minimum, there are recommendations for supplementation with folic acid, vitamin D, and iron. Any need for calcium supplement must be based on intake of dairy products. These micronutrients are discussed in more detail below and Table 5 shows recommendations.

### 6.1. Vitamin B9/Folic Acid

Folates are important vitamins in pregnancy. Folate is critical for the synthesis of nucleic acids and, thus, cell division, therefore being important in the foetal growth. If the maternal folate level is low, then the risk of low birth weight and neural tube defects increases. Supplementation with folic acid (the synthetic structure of the folate family) during the periconceptional period has been shown to reduce the risk of these outcomes in NGTP [81,82,83]. The IOM recommends a daily intake of 600 µg/d during pregnancy [8], while the Nordic Council of Ministers 2014 has a lower recommendation of 500 µg/d in pregnancy [11]. A daily supplement of 400 µg folic acid/d may be recommended for all women of childbearing age and during the first 12 week of gestation to avoid low levels of folate in the mother at conception and ensure sufficient dietary intake.

Of notice, the form of folate substitution might be relevant to take into consideration. Common genetic variations in the genes encoding proteins that are involved in folate metabolism can lead to a lower conversion rate of folate to the active form, L-methylfolate. Recently, focus has been put on supplementation with L-methylfolate rather than folic acid. Apparently, women with such genetic mutations may benefit from direct supplementation with L-methylfolate [84].

Some studies have found that homocysteine levels, which are a marker of low folate or vitamin B12 status, are higher in women with GDM as compared to non-diabetic pregnant women. As an example, a cross-sectional study conducted by Guven et al. showed a higher homocysteine concentration in second trimester. However, folate and vitamin B12 levels did not differ between groups [85] and, at present, the same recommendations as for NGTP apply to women with GDM.

### 6.2. 25-Hydroxyvitamin D

The IOM recommends a dietary intake of 5.0 µg vitamin D/d during pregnancy [8], while NNR, which covers the Nordic countries, where serum 25(OH)D concentrations are often low in winter, recommends 10 µg/d during pregnancy [11]. These recommendations for NGTP are also currently applicable to women with GDM.

Increasing evidence suggests that vitamin D may play an important role in modifying the risk of diabetes [86], as vitamin D acts directly on the pancreatic beta cell by increasing insulin secretion, and indirectly by attenuating systemic inflammation that is associated with insulin resistance [87,88]. Many cross-sectional and prospective observational studies have shown an inverse association between vitamin D status and the prevalence or incidence of type 2 diabetes [86]. Therefore, vitamin D is also the micronutrient that has been studied most extensively in relation to GDM. Several studies indicate a significant inverse relation of serum 25OHD and the incidence of GDM, but it is not clear whether this association is causal [89] and large RCTs of the effects of vitamin D in women with GDM are sparse. However, in a RCT by Asemi et al., 54 women with GDM received either placebo capsules or vitamin D capsules (50.000 IU) twice during the six week study period and intake of vitamin D supplements led to a significant decrease in FPG and insulin resistance assessed by HOMA-IR [90]. In another RCT, women with GDM were randomized to either placebo or 200 IU, 2000 IU, or 4000 IU vitamin D daily. Insulin levels, HOMA-IR, and total cholesterol were significantly reduced in the group receiving 4000 IU of vitamin D [91]. In a recent meta-analysis, including six RCTs, it was found that vitamin D supplementations improved insulin resistance and LDL cholesterol, but had no beneficial effect on FPG, insulin, HbA1c, total-, HDL-cholesterol, and triglycerides concentrations [92].

The effects of vitamin D supplementation in GDM are equivocal and the available trials have been conducted in different settings with differences in subject populations, length of intervention, and forms of vitamin D supplementation. Confounding variables, such as ethnicity and seasonality, add to the complexity of vitamin D studies and vitamin D can be seen as a proxy for a healthy lifestyle with an active life outside being exposed to the sun. At present, it is therefore difficult to conclude whether vitamin D can reduce the risk of developing GDM and/or improve glycaemic control in women with GDM and vitamin D deficiency/insufficiency, as there is a need for larger well-designed RCTs that evaluate interventions together with the evaluation of confounding factors.

### 6.3. Calcium

The requirement of calcium is increased during pregnancy [93]. However, the Nordic Council of Ministers 2014 did not find enough data to draw firm conclusions on potential association between calcium intake during pregnancy and bone health in the offspring. The recommended daily intake of 900 mg/day was kept unchanged from the 2004 to the 2012 updated version [11]. However, the IOM has a slightly higher recommendation during pregnancy of 1000 mg/day in women >19 years [8].

Whether supplementation is necessary depends on the woman’s food intake. However, calcium supplementation might have a potential positive effect on glycaemic control in women with GDM. Asemi et al. demonstrated a significant reduction in FPG in women with GDM who received 1000 mg calcium/d plus 50.000 U vitamin D3 supplements twice during a six week intervention when compared to placebo. In the same study, Asemi et al. also found a significant reduction in the serum insulin levels and HOMA-IR. It was concluded that calcium plus vitamin D supplementation in women with GDM had beneficial effects on their metabolic profile [93].

In conclusion, it can be advocated to ensure a minimum intake of 900–1000 mg calcium per day during pregnancy in women with GDM. Therefore, it can be recommended that all pregnant women receive e.g., 0.5 L of milk product per day, less when supplemented with cheese, or that 900–1000 mg calcium is ingested daily from other sources of calcium. If the woman is unable to meet these recommendations, then there may be a need of a daily supplement of 500 mg of calcium throughout pregnancy.

### 6.4. Iron

Iron deficiency is the most common micronutrient deficiency in pregnancy and during childbearing years. Women have increased needs for iron due to the iron losses during menstrual bleeding [11]. Additionally, many women have small iron stores, when they become pregnant and are not gaining appropriate amounts of iron in their diet to cover the increased need during pregnancy. Because of this, some countries recommend iron supplements of 40 mg as early as week 10 of pregnancy [94]. Maternal iron need increases during pregnancy in order to accommodate the growth and maintenance of the foetus and uterus and the increased red blood cell count. Further, there is an expected iron loss when giving birth [11]. The IOM recommends a daily intake of 27 mg/d during pregnancy [8], while iron supplementation of 40 mg/d from week 18–20 of gestation has been suggested by the Nordic Council of Ministers 2014, in order to reduce the risk of low birth weight and preterm delivery [11,95].

However, whether iron supplementation during pregnancy is necessary or a toxic supplement is a controversial topic. The literature suggests that iron influences glucose metabolism [95]. In a cohort study conducted by Bo et al., an association between the intake of iron supplements and a higher oral glucose tolerance test glucose values in women with GDM was found [95]. Today, there is not enough evidence to suggest a different recommendation for iron intake in women with GDM than what applies to NGTP.

## 7. Probiotics

In recent years, the role of gut microbiota in regulating metabolism has become a hot topic of investigation. Thus, gut microbiota may play a significant role in the development of obesity and may also have an important impact on glucose homeostasis [96]. Moreover, the results indicate that, in pregnancy, the changes in gut microbiota from the first to the third trimester may contribute to the maternal metabolic changes [97]. In a Danish study, the gut microbiota profiles were investigated in 50 women with GDM and in 157 pregnant women with normal glucose tolerance and it was reported that, in the third trimester of pregnancy, GDM was associated with an altered gut microbiota as compared to that of NGTP [98]. Accordingly, several studies have been performed to determine whether probiotics could be beneficial for the prevention or treatment of GDM. However, the results of the many available studies are equivocal. In a Finnish RCT study, 439 pregnant women with overweight or obesity were divided into four intervention groups with fish oil + placebo, probiotics (*Lactobacillus rhamnosus* and *Bifidobacterium animalis* ssp *lactis*) + placebo, fish oil + probiotics, and placebo + placebo. The primary outcomes were incidence of GDM and change in fasting glucose in the intervention period, but no benefits in lowering the risk of GDM or improving glucose metabolism was found in any of the groups [99]. Callaway et al. performed a large double-blind RCT, including 411 women, in order to determine whether probiotics (*Lactobacillus rhamnosus* and *Bifidobacterium animalis* ssp *lactis*) that were administered from the second trimester in women with overweight or obesity could prevent GDM. Unfortunately, GDM could not be prevented by the intervention [100]. In an Irish RCT, 149 women with GDM received either a probiotic capsule (*Lactobacillus salivarius*) or placebo once daily from diagnosis of GDM to delivery and no effect on glycaemic control was found [101].

However, two meta-analyses have shown that the use of probiotics was associated with an improved glucose and lipid metabolism in pregnant women, and could tentatively reduce the risk of gestational diabetes [102,103]. Another meta-analysis showed that supplementation with probiotic reduced insulin resistance (HOMA-IR) and fasting serum insulin in women with gestational diabetes significantly, as compared to pregnant women with normal glucose tolerance [104]. In a recent study conducted by Kijmanawat et al., women with GDM were randomized to probiotics (*Lactobacillus* and *Bifidobacterium*) or placebo for four consecutive weeks and a significant improvement in glucose metabolism in the probiotic group, regarding fasting glucose, insulin, and HOMA-IR was found [105]. Additionally, in a study conducted by Karamali et al., where 60 women with GDM were included to determine the effects of probiotic supplementation on glycaemic control and lipid profiles after six weeks and beneficial effects on glycaemic control, triglycerides, and VLDL cholesterol were reported. The study was a double blind RCT where the women either received a probiotic capsule (containing three viable freeze-dried strains: *Lactobacillus acidophilus*, *L. casei*, and *Bifidobacterium bifidum*) or a matching placebo [106].

### Summary, Probiotics

The question of whether gut microbiota modification could be an effective tool in improving glycemic control and reducing insulin resistance in pregnant women with GDM is complicated. The results differ as the human gut houses a complex microbial ecosystem and the present studies have used different pre-or probiotics or multi-strain probiotics, making it difficult to compare studies and to make a final conclusion at this point.

## 8. Nutrition Counselling

In a recent meta-analysis, including 18 RCTs involving 1151 women with GDM, a moderate effect of dietary interventions on maternal glycaemic outcomes, including changes in fasting, post-breakfast, and postprandial glucose levels, and the need for medication treatment was found [6]. For neonatal outcomes, including 16 RCTs and 841 women with GDM, it was found that modified dietary interventions were associated with lower infant birth weight and less macrosomia [6]. These associations were found despite a high heterogeneity between studies [6], which indicated that several methods can be used and the dietary guidance should probably be adapted to the individual patient.

The American Diabetes Association recommends that women with GDM receive an individualized nutrition plan as a part of medical nutrition therapy. The nutrition plan should be developed in collaboration between the women and an experienced dietician [10]. The adjustment of the nutrition plan should be continuous and based upon self-glucose monitoring, appetite, and weight-gain patterns, as well as consideration for maternal dietary preferences and work, leisure, and exercise. If insulin therapy is added to nutrition therapy, a primary goal is to maintain carbohydrate consistency at meals and snacks in order to facilitate insulin adjustment.

## 9. Physical Activity

In non-pregnant individuals, it is well established that physical activity reduces insulin resistance by stimulating the glucose transporters on the surface of skeletal muscle cells and thereby improving glucose uptake [107,108,109]. Interestingly, whereas many studies have addressed the impact of physical activity on various outcomes in pregnancy in general, only a paucity of studies have addressed the impact of physical activity on maternal blood glucose levels and glycaemic control during pregnancy in women with GDM.

### 9.1. Short Term Effects of Physical Activity in Pregnancy on Maternal Blood Glucose Levels

Acute bouts of physical activity appear to influence maternal glucose levels on short term. Treadmill exercise for 30 min reduces blood glucose and insulin levels in healthy pregnant women [110]. Among women at risk of GDM 20 min of moderate intensity cycling after an oral glucose tolerance test reduced blood glucose excursions and insulin levels within one to two hours after glucose ingestion [111]. However, a long-term effect was not observed, when evaluating continuous glucose measurements for up to 48 h after physical activity [111]. Similar findings were observed after walking, i.e., women at risk of GDM had decreases in blood glucose levels that were associated with the duration and intensity of the exercise with glucose levels aligning within a few hours after physical activity [112].

Similar observations have been made among women with GDM. Light intensity walking after a meal reduced 1-h blood glucose levels, but not 2-h values [113]. Moderate intensity walking after a meal had slightly longer lasting effects on blood glucose levels with effects visible for two to three hours where after blood glucose levels again aligned [114]. Cycling at mild and moderate intensity yielded similar results as after walking, i.e., a short-lasting decreasing effect on blood glucose levels when compared to the resting condition in a “dose-dependent” matter, i.e., larger effects with more intensified physical activity [115].

In the above-mentioned studies, blood glucose levels after physical activity were comparable after minutes to hours. Thus, is appears reasonable that acute bouts of physical activity have short lasting effects on maternal glucose levels. A continuous program of physical activity appears to be necessary for longer-term effects to be seen.

### 9.2. Longer-Term Effects of Physical Activity

Longer-term effects of bouts of physical activity are more diverse, as the effects could be the direct influence upon glucose metabolism or it could be effects relating to pregnancy outcomes for which glucose metabolism plays a role, i.e., birth weight and a range of pregnancy complications, such as hypertensive disorders, macrosomia, shoulder dystocia, and neonatal hypoglycaemia and jaundice.

Resistance exercise has been reported to be effective in reducing the need for insulin in GDM pregnancy [116], and moderate intensity cycling three times weekly in combination with diet was able to yield weekly blood glucose levels that were comparable to insulin combined with diet [117]. Again, exercising women managed to stay without any need for insulin [117]. In contrast, combined cycling exercise at moderate intensity alternated by walking three to four times weekly did not induce changes in daily blood glucose measurements or in HbA1c values [118].

The effects of physical exercise during GDM pregnancy on pregnancy outcomes have not been thoroughly examined. Often, study protocols have combined physical activity with other lifestyle modifications, so that the individual contributions from diet, physical activity, coaching, or other included interventions on the study outcomes may be difficult to discern. In a 2018 Cochrane overview of reviews, it was concluded that, in general, only limited effects of exercise as the sole intervention in GDM pregnancy could be documented. Of the palette of interventions that could be explored, the best documentation was available for the combination of healthy eating, physical exercise, and self-monitoring of blood glucose levels. In combination, these efforts could reduce the risk of LGA-babies, but probably at the cost of more prevalent inductions of labour [119]. Thus, the beneficial effects of lifestyle interventions in pregnancy could be accompanied by an introduction of side effects or potential harms in pregnancy [119].

### 9.3. Recommendations for Exercise in GDM Pregnancy

In Denmark, pregnant women are recommended at least 30 min of (unspecified) moderate intensity physical exercise daily. There are no specific recommendations for physical activity or exercise that addresses women with GDM, but women with GDM are encouraged to exercise more than the recommendations in NGTP [120]. Similar recommendations are found in the Canadian guidelines for physical activity throughout pregnancy [121], in which 150 min of moderate intensity physical activity each week on at least three separate days is recommended for women independent of GDM status.

Exercise three times a week for 40 to 60 min at 65 to 75% of the age-corrected heart rate maximum has been suggested for women with GDM [122]. Activities could be circuit training, walking, or cycling, but the need for studies testing the most optimal physical activity was acknowledged [122].

Thus, physical activity during pregnancies complicated with GDM is recommended, and moderate intensity activity appears to be the choice agreed upon. However, currently, there is no common agreement on the type, frequency, and duration of physical activity that would be beneficial or even most optimal. Further, the optimal gestational age or the optimal range of gestational weeks for intervention needs to be clarified.

### 9.4. Societal Interventions

The increased prevalence of diabetes mellitus in especially industrialized countries have led to considerations regarding possible societal interventions. The construction of urban environments aimed at facilitating physical activity has been considered. Easy access to minor and local sport facilities might be an opportunity to improve physical activity for some individuals; however, this strategy is dependent on whether the individuals will use such facilities. Urban planning may be a means to increase the level of physical activity on a population level, and it has been reported that increasing the “walkability” of a neighbourhood is associated with a lower incidence of diabetes [123]. Walking has been suggested to be an especially attractive means of physical activity during pregnancy [124]. In GDM, a single study recently reported on the relationship between neighbourhood walkability and variables that were related to GDM [125]. High neighbourhood walkability was, in general, associated to a lower pre-pregnant BMI and higher pre-pregnant levels of physical activity. In pregnancy, though, increasing walkability of neighbour surroundings was not associated to GWG, insulin sensitivity, glycaemia, or beta cell function [125] Additionally, no difference in GDM prevalence was observed across the different classes of walkable surroundings [125].

Despite low evidence for the time being of the effect of walking on the risk for GDM in pregnancy, walking that is facilitated on both the individual and societal levels may prove to be a simple and obtainable way to introduce more physical energy expenditure in pregnancy [124,125].

### 9.5. Hindrances to Exercise in Pregnancy

During pregnancy, certain conditions may limit physical activity. Pre-existing medical conditions may limit the amount of physical activity that can be performed. Musculoskeletal or cardiac diseases may decrease the daily level of physical activity and preclude any invigorated physical activity. Additionally, conditions that are related to pregnancy may lead to the recommendation of immobilization or even bed rest, e.g., short cervix conditions or imminent premature delivery. Despite the lack of evidence for promoting immobilization of women with such complications, clinical practice implies that some degree of immobilization is often instituted. In the case of threatening preterm delivery, the administration of corticoid therapy for foetal lung maturation may further exacerbate insulin resistance, at least for days [126]. Furthermore, common conditions, like pelvic joint laxity and pelvic girdle discomfort, will often lead to cautious movements and decreased levels of physical activity. More uncommon, lower extremity varicose veins or even deep venous thrombosis may cause immobilization. Such conditions are primarily related to the third trimester of pregnancy, i.e., at the time of maximal insulin resistance.

### 9.6. Summary, Physical Activity

When GDM is present, single physical activities clearly has short term effects on blood glucose levels. However, sustainable effects are more complex to obtain. Long-lasting effects, be it on maternal blood glucose levels or on pregnancy outcomes in general, do with all likelihood depend on daily physical activity and may be further corroborated by a concomitant reduction in GWG. Measures to increase the daily level of physical activity and the strategy for exercise and physical activity in pregnancy with GDM still need further exploration.

## 10. Conclusions

A summary of the above recommendations is found in Table 6. All women with GDM should be offered dietary advice by a clinical dietitian, as dietary counselling the cornerstone in the treatment of GDM. Knowledge of the impact of diet on blood glucose is of great importance in preventing complications, such as birth complications, caesarean section, LGA-babies, and type 2 diabetes, later in life. The woman should receive guidance on how to construct a varied diet and how to avoid hyperglycaemia. Particular efforts should focus on carbohydrate intake as both type, amount and distribution of carbohydrate are of major importance for the postprandial blood glucose. In general, the same recommendations for minerals and vitamins apply to women with GDM as in NGTP. In addition, physical activity of moderate intensity for at least 30 min daily or 150 min weekly should be encouraged, as this may contribute to improved glycaemic control.

## Figures and Tables

**Figure 1 nutrients-12-03050-f001:**
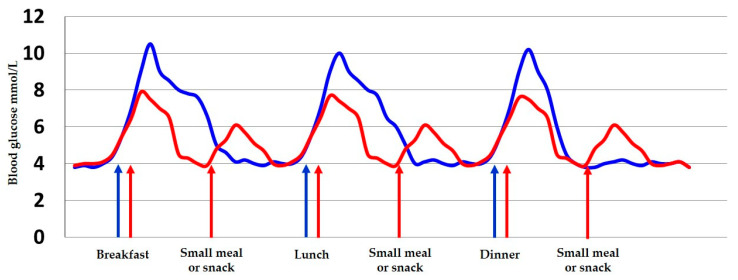
The blood glucose levels according to different strategies for daily food intake. Blue curve illustrates the normal meal pattern and red curve illustrates meal pattern in women with gestational diabetes mellitus (GDM) to avoid excessive blood glucose fluctuations and to preserve the planned number of calories to be ingested. Blue arrows: Three main meals. Red arrows: three main meals and three snacks.

**Table 1 nutrients-12-03050-t001:** Recommendations for total weight gain during singleton pregnancy.

Pre-Pregnancy BMI	Total Weight Gain (Range in kg)
Underweight (<18.5 kg/m^2^)	12.5–C18
Normal weight (18.5–24.9 kg/m^2^)	11.5–16
Overweight (25.0–29.9 kg/m^2^)	7–11.5
Obese(≥30 kg/m^2^)	5–9

Modified from Table S1 in the IOM report by Rasmussen & Yaktine, “Weight Gain During Pregnancy: Reexamining the Guidelines (2009)” [8]. BMI, body mass index.

**Table 2 nutrients-12-03050-t002:** Equations to calculate estimated energy requirement for nonpregnant women.

	NNR		IOM
Age	MJ/d	Age	kcal/d
11–18	(0.0393 W + 1.04 H + 1.93)*PAL	14–18	135.3 − (30.8 × age [y]) + PA × [(10.0 × weight [kg]) + (934 × height [m])] + 25
19–30	(0.0546 W + 2.33)*PAL	>19	354 − (6.91 × age [y]) + PA × [(9.36 × weight [kg]) + (726 v height [m])]
31–60	(0.0433 W + 2.57 H − 1.180)*PAL		

NNR, Nordic Nutrition Recommendations; IOM, Institute of Medicine; PA, physical activity coefficient; PAL, physical activity level; MJ, mega Joule; W, weight in kilograms; H, height in meters, d, day. Modified from the IOM report by Rasmussen & Yaktine 2009, “Weight Gain During Pregnancy: Reexamining the Guidelines” and The Nordic Council of Ministers 2014 “Nordic Nutrition Recommendations: Integrating nutrition and physical activity” [8,11].

**Table 3 nutrients-12-03050-t003:** Physical activity level (PAL) for use in equations for energy requirement recommended by NNR and Physical Activity Coefficients (PA values) for use in equations for Energy requirement recommended by IOM.

**PAL**	**NNR**
1.1–1.2	Bed-bound or chair-bound
1.3–1.5	Seated work with none or only little physical activity
1.6–1.7	Seated work with some movement or some physical activity
1.8–1.9	Work including standing and moving around or seated work with some movement and with frequent activity
2.0–2.4	Very strenuous work or daily competitive physical training
**PA, age ≥19 (ages 14–18)**	**IOM**
1.0 (1.0)	Very low active level
1.12 (1.16)	Low active level
1.27 (1.31)	Active level
1.45 (1.56)	Highly active level

IOM, Institute of Medicine; NNR, Nordic Nutrition Recommendations; PA, physical activity coefficient; PAL, physical activity level. Modified from Table 8.7 chapter 8 in the Nordic Council of Ministers 2014 guideline “Nordic Nutrition Recommendations: Integrating nutrition and physical activity” [8] and Table B-1C from the IOM report by Rasmussen & Yaktine, “Weight Gain During Pregnancy: Reexamining the Guidelines (2009)” [11].

**Table 4 nutrients-12-03050-t004:** Additional daily calorie requirements during pregnancy.

Trimester	NNR	IOM
1st trimester	103 kcal	0 kcal
2nd trimester	329 kcal	340 kcal
3rd trimester	537 kcal	452 kcal

IOM, Institute of Medicine; NNR, Nordic Nutrition Recommendations [8,11].

**Table 5 nutrients-12-03050-t005:** Recommendation of specific micronutrients in pregnancy.

Micronutrient	NNR	IOM
Folic acid, µg/day	500	600
25-Hydroxyvitamin D, µg/day	10	5
Calcium, mg/day	900	1000
Iron, mg/day	40	27

IOM, Institute of Medicine; NNR, Nordic Nutrition Recommendations [8,11].

**Table 6 nutrients-12-03050-t006:** Summary of recommendations.

Dietary Components	Recommendations
Energy	Excessive weight gain should be avoided and a calorie restriction of 30–33% is advisable in women with overweight or women who have already gained the recommended weight during pregnancy
Carbohydrates	Exact amount of carbohydrate should be individualized. A minimum of 175 g/d should be ensured. Patients should be guided to choose starchy foods such as vegetables, legumes, fruits, and whole grains.Carbohydrate intake should be distributed throughout the day.
Protein	Total amount of protein should be 10–35E% with a minimum of 71 g/d. Protein intake should primarily come from plants, lean meat, and fish.
Fat	Total amount of fat should be 20–40E% with a maximum of 10E% from saturated fat, a minimum of 10–20E% from MUFAs, and 5–10E% from PUFAs. An intake of a minimum 350g of fish/week may be advisable.
Folic acid	500–600 µg/d is recommended. Daily supplement of 400 µg/d may be advisable for all women at childbearing age and during the first 12 week of gestation.
25-Hydroxyvitamin D	5–10 µg/d is recommended depending on how much sunlight the woman gets.
Calcium	900–1000 mg/d is recommended. Supplement may be advisable in women with a lack of intake of dairy products.
Iron	27–40 mg/d is recommended.
Probiotics	It remains unresolved whether probiotics have beneficial metabolic effects in women with GDM.

d, daily; E%, energy precent; GDM, gestational diabetes mellitus; MUFAs, monounsaturated fatty acids; PUFAs, polyunsaturated fatty acids.

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
