# Peer review of "Diet and Healthy Lifestyle in the Management of Gestational Diabetes Mellitus"

_nutrients, 2020, doi:10.3390/nu12103050_

Round 1

Reviewer 1 Report

Manuscript “Diet and healthy lifestyle in the management of GDM”

 The objective of the manuscript was to present first-line treatment in GDM: nutrition therapy and physical activity. The topic of this work is not innovative but extremely important from a clinical point of view.  In this regard, the manuscript is very interesting and pertinent.

However, there are some aspects of this article that could be improved:

  • Abstract-lines 20-21: “Hyperglycaemia is primarily dependent on carbohydrate-intake, why nutrition counseling…” - the construction of this sentence is unclear and requires some rearrangement
  • Introduction- line 34:  Not “fetal” but foetal insulin production. This requires standardization throughout the work.
  • Introduction- line 40: “Lifestyle changes are essential in both the management…”-
    please correct this sentence
  • Energy requirements - line 98: What do you mean by : “healthier foetal growth” ?
  • 2 Lines: 62 and 90 :In both sub-headings “in” is redundant
  • 2 Line 100 - please correct the sentence
  • Carbohydrates-line 109: “a rise in blood glucose” – it should be better to say: a rise in blood glucose level
  • 1 Low-Carbohydrate Diets – line 129 – not “lower Area” but lower area
  • 6 Summary, Carbohydrates – lines 323-233: the structure of the sentence starting with “Postprandial hypergycaemia...” needs to be improved
  • Protein-lines 242-243: the sentence starting with “During pregnancy..” needs to be improved.
  • 2 Protein, the placenta and GDM : line 269: “birthweight” – it should be written separately: birth weight
  • 2 Protein, the placenta and GDM : line 272: “the placenta?” - a question mark is redundant here
  • 2 Protein, the placenta and GDM : line 279: “birthweight” – in the opinion of the reviewer it should be written separately: birth weight. The same mistake is in line 294 and throughout the rest of the manuscript
  • 2. Long-term effects of physical activity: in line 591 should be shoulder dystocia
  • Conclusion- line 670: “The women should receive guidance on how to construct a varied diet and to hyperglycaemia” –sentence construction needs to be corrected, for example: The women should receive guidance on how to construct a varied diet and how to avoid hyperglycaemia.

The work requires checking by a native-speaker.

Reviewer 2 Report

The topic of the manuscript is relevant and important. The authors managed to provide an nicely structured overview of the available evidence in a narrative review. However, certain remarks should be adressed.

Major remarks:

  • The use of fetus or foetus should be consistent throughout the manuscript. Currently this is not the case (e.g. line 34: fetal insuline production vs. foetal growth).
  • The text would benefit from a revision of the English language. In the current state, the flow and readability are hampered with due to errors in syntax and grammar.
  • Attention should be given to the use of people-first language: e.g. 'women with obesity' instead of 'obese women'.
  • It might be of interest to add a paragraph to the manuscript regarding rate of gestational weight gain. Recommended weight gain per trimester is relevant to clinicians and can permit earlier interventions in patients with excessive gestational weight gain, before IOM guidelines are exceeded.
  • Line 419: please note that the form of folate substitution might be relevant. Common genetic polymorphisms in methyl tetrahydrofolate reductase can lead to a lower conversion rate of THF to the active form 5-MTHF.
  • Line 665: a table with an overview of the recommendations made in this article would promote usability in the clinical setting. 
  • Did the author's find any reference in the collected literature to the prevention of the fetal glucose steal phenomenon?

Minor remarks:

  • Please add the abbrevation for PAL to the legend of table 2 (line 69).
  • Line 90: do you mean 'energy requirements for women with overweight', or 'energy requirements for women with excessive gestational weight gain'? 
  • Line 177: can you please elaborate on the complications that were less prelevant?
  • Line 272: remove unnecessary '?'
  • Line 434: a reference to the DALI trial on vitamin D supplementation in the prevention of GDM should be considered.
  • Line 651: there is insufficient evidence to promote immobilization in patients with preterm labour or cervical shortening. 

Round 2

Reviewer 2 Report

A good effort has been done by the authors to improve the quality of the paper. However, some minor remarks remain: 

  • English language and style have been greatly improved throughout the manuscript. Still, certain paragraphs, mainly those which have been added due to suggestions of the reviewers, continue to suffer from errors in grammar and syntax. E.g. lines 60 - 65: "Women, who are underweight (BMI<18.5 kg/m2), should be recommended a weight gain between 0.44-0.58 kg/week, women, who has a normal weight (BMI=18.5-24.9 kg/m2); 0.35-0.50 kg/week, women, who are overweight (BMI=25.0-29.9 kg/m2); 0.23-0.33 kg/week and finally women, who are obese (BMI≥30kg/m2), should be recommended a weight gain between 0.17-0.27 kg/week during 2nd and 3rd trimester."
  • Line 112: "the women with GDM, who had...". What follows is an essential clause to these women. The comma should be ommited.
  • Line 123: "carbohydrates are..." in stead of "carbohydrate is..."
  • Line 134: "The IOM recommends limiting the intake of ..."
  • Line 156: "Normally" is used twice in one sentence.
  • Line 272: "In early GDM, when patients have less metabolic decompensation,..."
  • Line 437: The sentence on the form of folate substitution is very vague. In its current form, it lacks meaning and a clear message. 
  • Line 448 and onwards: "25(OH)D" instead of 25OHD.
  • Line 501: "Foetus" instead of "fetus".
  • Line 523: people-first language should be used. "439 pregnant women with overweight or obesity"
  • Line 530: "in women with overweight or obesity"
  • Line 639: the past tense of the verb 'lead' is led
  • Table 6: "women who have already gained..."
  • Line 682: "a summary of the above recommendations is found in..."
